# A New Distribution on the Simplex with Auto-Encoding Applications

**Andrew Stirn**,[*] **Tony Jebara**,[†] **David A Knowles**[‡]
Department of Computer Science
Columbia University
New York, NY 10027
`{andrew.stirn,jebara,daknowles}@cs.columbia.edu`

## Abstract

We construct a new distribution for the simplex using the Kumaraswamy distribution and an ordered stick-breaking process. We explore and develop the theoretical properties of this new distribution and prove that it exhibits symmetry (exchangeability) under the same conditions as the well-known Dirichlet. Like the Dirichlet, the new distribution is adept at capturing sparsity but, unlike the Dirichlet, has an exact and closed form reparameterization–making it well suited for deep variational Bayesian modeling. We demonstrate the distribution's utility in a variety of semi-supervised auto-encoding tasks. In all cases, the resulting models achieve competitive performance commensurate with their simplicity, use of explicit probability models, and abstinence from adversarial training.

## 1 Introduction

The Variational Auto-Encoder (VAE) [12] is a computationally efficient approach for performing variational inference [11, 27] since it avoids per-data-point variational parameters through the use of an inference network with shared global parameters. For models where stochastic gradient variational Bayes requires Monte Carlo estimates in lieu of closed-form expectations, [23, 12] note that low-variance estimators can be calculated from gradients of samples with respect to the variational parameters that describe their generating densities. In the case of the normal distribution, such gradients are straightforward to obtain via an explicit, tractable reparameterization, which is often referred to as the "reparameterization trick". Unfortunately, most distributions do not admit such a convenient reparameterization. Computing low-bias and low-variance stochastic gradients is an active area of research with a detailed breakdown of current methods presented in [4]. Of particular interest in Bayesian modeling is the well-known Dirichlet distribution that often serves as a conjugate prior for latent categorical variables. Perhaps the most desirable property of a Dirichlet prior is its ability to induce sparsity by concentrating mass towards the corners of the simplex. In this work, we develop a surrogate distribution for the Dirichlet that offers explicit, tractable reparameterization, the ability to capture sparsity, and has barycentric symmetry (exchangeability) properties equivalent to the Dirichlet.

Generative processes can be used to infer missing class labels in semi-supervised learning. The first VAE-based method that used deep generative models for semi-supervised learning derived two variational objectives for the same the generative process–one for when labels are observed and one for when labels are latent–that are jointly optimized [13]. As they note, however, the variational distribution over class labels appears only in the objective for unlabeled data. Its absence from

---

[*]jointly affiliated with New York Genome Center
[†]jointly affiliated with Spotify Technology S.A.
[‡]jointly affiliated with Columbia University's Data Science Institute and the New York Genome Center

the labeled-data objective, as they point out, results from their lack of a (Dirichlet) prior on the (latent) labels. We suspect they neglected to specify this prior, because, at the time, it would have rendered inference intractable. They ameliorate this shortcoming by introducing a discriminative third objective, the cross-entropy of the variational distribution over class labels, which they compute over the labeled data. They then jointly optimize the two variational objectives after adding a scaled version of the cross-entropy term. Our work builds on [13], while offering some key improvements. First, we remove the need for adding an additional discriminative loss through our use of a Dirichlet prior. We overcome intractability using our proposed distribution as an approximation for the Dirichlet posterior. Naturally, our generative process is slightly different, but it allows us to consider only unmodified variational objectives. Second, we do not stack models together. Kingma et al.'s best results utilized a standard VAE (M1) to learn a latent space upon which their semi-supervised VAE (M2) was fit. For SVHN data, they perform dimensionality reduction with PCA prior to fitting M1. We abandon the stacked-model approach in favor of training a single model with more expressive recognition and generative networks. Also, we use minimal preprocessing (rescaling pixel intensities to $[0, 1]$).

Use of the Kumaraswamy distribution [14] by the machine learning community has only occurred in the last few years. It has been used to fit Gaussian Mixture Models, for which a Dirichlet prior is part of the generative process, with VAEs [19]. To sample mixture weights from the variational posterior, they recognize they can decompose a Dirichlet into its stick-breaking Beta distributions and approximate them with the Kumaraswamy distribution. We too employ the same stick-breaking decomposition coupled with Kumaraswamy approximations. However, we improve on this technique by expounding and resolving the order-dependence their approximation incurs. As we detail in section 2, using the Kumaraswamy for stick-breaking is not order agnostic (exchangeable); the generated variable has a density that depends on ordering. We leverage the observation that one can permute a Dirichlet's parameters, perform the stick-breaking sampling procedure with Beta distributions, and undo the permutation on the sampled variable without affecting its density. Those same authors also use this Beta-Kumaraswamy stick-breaking approximation to fit a Bayesian non-parametric model with a VAE [20]. Here too, they do not account for the impact ordering has on their approximation. Their latent space, being non-parametric, grows in dimensions when it insufficiently represents the data. As we demonstrate in section 2.2 and fig. 1, approximating sparse Dirichlet samples with the Kumaraswamy stick-breaking decomposition without accounting for the ordering dependence produces a large bias in the samples' last dimension. We conjecture that their Bayesian non-parametric model would utilize fewer dimensions with our proposed distribution and would be an interesting follow up to our work.

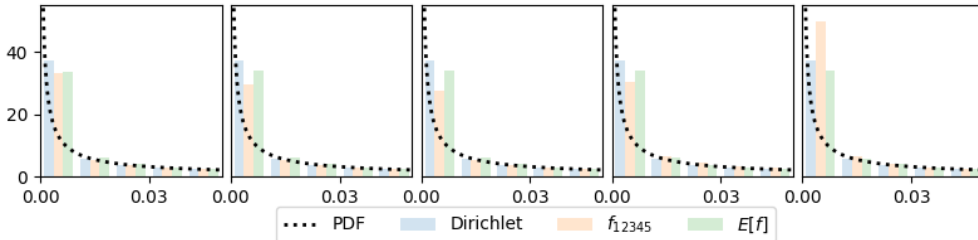

Figure 1: Sampling bias for a 5-dimensional sparsity-inducing Dirichlet approximation using $\alpha = \frac{1}{5}(1, 1, 1, 1, 1)$. We maintain histograms for each sample dimension for three methods: Dirichlet, Kumaraswamy stick-breaks with a fixed order, Kumaraswamy stick-breaks with a random ordering. Note the bias on the last dimension when using a fixed order. Randomizing order eliminates this bias.

## 2 A New Distribution on the Simplex

The stick-breaking process is a sampling procedure used to generate a $K$ dimensional random variable in the $K-1$ simplex. The process requires sampling from $K-1$ (often out of $K$) distributions each with support over $[0, 1]$. Let $p_i(v; a_i, b_i)$ be some distribution for $v \in [0, 1]$ parameterized by $a_i$ and $b_i$. Let $o$ be some ordering (permutation) of $\{1, \ldots, K\}$. Then, algorithm 1 captures a generalized stick-breaking process. The necessity for incorporating ordering will become clear in section 2.1.

---

**Algorithm 1** A Generalized Stick-Breaking Process

---

**Require:** $K \geq 2$, base distributions $p_i(v; a_i, b_i) \ \forall \ i \in \{1, \ldots, K\}$, and some ordering $o$
    Sample: $v_{o_1} \sim p_{o_1}(v; a_{o_1}, b_{o_1})$
    Assign: $x_{o_1} \leftarrow v_{o_1}, i \leftarrow 2$
    **while** $i < K$ **do**
        Sample: $v_{o_i} \sim p_{o_i}(v; a_{o_i}, b_{o_i})$
        Assign: $x_{o_i} \leftarrow v_{o_i}\left(1 - \sum_{j=1}^{i-1} x_{o_j}\right), i \leftarrow i + 1$
    **end while**
    Assign: $x_{o_K} \leftarrow 1 - \sum_{j=1}^{K-1} x_{o_j}$
    **return** $x$

---

From a probabilistic perspective, algorithm 1 recursively creates a joint distribution $p(x_{o_1}, \ldots, x_{o_{K-1}})$ from its chain-rule factors $p(x_{o_1})p(x_{o_2}|x_{o_1})p(x_{o_3}|x_{o_2}, x_{o_1}) \ldots p(x_{o_{K-1}}|x_{o_{K-2}}, \ldots x_{o_1})$. Note, however, that $x_{o_K}$ does not appear in the distribution. Its absence occurs because it is deterministic given $x_{o_1}, \ldots, x_{o_{K-1}}$ (the $K - 1$ degrees of freedom for the $K - 1$ simplex). Each iteration of the **while** loop generates $p(x_{o_i}|x_{o_{i-1}}, \ldots, x_{o_1})$ by sampling $p_{o_i}(v; a_{o_i}, b_{o_i})$ and a change-of-variables transform $T_i : [0, 1]^i \rightarrow [0, 1]^i$ to the samples collected thus far. This transform and its inverse are

$$T_i(x_{o_1}, \ldots, x_{o_{i-1}}, v_{o_i}) = \left(x_{o_1}, \ldots, x_{o_{i-1}}, v_{o_i}\left(1 - \sum_{j=1}^{i-1} x_{o_j}\right)\right) \tag{1}$$

$$T_i^{-1}(x_{o_1}, \ldots, x_{o_{i-1}}, x_{o_i}) = \left(x_{o_1}, \ldots, x_{o_{i-1}}, x_{o_i}\left(1 - \sum_{j=1}^{i-1} x_{o_j}\right)^{-1}\right). \tag{2}$$

Applying the change-of-variables formula to the conditional distribution generated by a **while** loop iteration, allows us to formulate the conditional as an expression involving just $p_i(v; a_i, b_i)$, which we assume access to, and $\det(J_{T_i^{-1}})$, where $J_{T_i^{-1}}$ is the Jacobian of eq. (2).

$$p(x_{o_i}|x_{o_{i-1}}, \ldots, x_{o_1}) = p(v_{o_i}|x_{o_{i-1}}, \ldots, x_{o_1}) \cdot \det(J_{T_i^{-1}}) = p_{o_i}(v; a_{o_i}, b_{o_i}) \cdot \left(1 - \sum_{j=1}^{i-1} x_{o_j}\right)^{-1}$$

A common application of the stick-breaking process is to construct a Dirichlet sample from Beta samples. If we wish to sample from Dirichlet$(x; \alpha)$, with $\alpha \in \mathbb{R}_{++}^K$, it suffices to assign $p_i(v; a_i, b_i) \equiv \text{Beta}(x; \alpha_i, \sum_{j=i+1}^{K} \alpha_j)$. With this assignment, algorithm 1 will return a Dirichlet distributed $x$ with density

$$p(x_{o_1}, \ldots, x_{o_K}; \alpha) = \frac{\Gamma\left(\sum_{i=1}^{K} \alpha_{o_i}\right)}{\prod_{i=1}^{K} \Gamma(\alpha_{o_i})} \prod_{i=1}^{K} x_{o_i}^{\alpha_{o_i} - 1}.$$

This form requires substituting for algorithm 1's final assignment $x_{o_K} \equiv 1 - \sum_{i=1}^{K-1} x_{o_K}$. Upon inspection, the Dirichlet distribution is order agnostic (exchangeable). In other words, given any ordering $o$, the random variable returned from algorithm 1 can be permuted to $(x_1, \ldots, x_K)$ (along with the parameters) without modifying its probability density. This convenience arises from the Beta distribution's form.

**Theorem 1** *For $K \geq 2$ and $p_i(v; a_i, b_i) \equiv Beta(x; \alpha_i, \sum_{j=i+1}^{K} \alpha_j)$, algorithm 1 returns a random variable whose density is captured via the Dirichlet distribution.*

A proof of theorem 1 appears in section 7.1 (appendix). A variation of this proof also appears in [5].

## 2.1 The Kumaraswamy distribution

The Kumaraswamy$(a, b)$ [14], a Beta-like distribution, has two parameters $a, b > 0$ and support for $x \in [0, 1]$ with PDF $f(x; a, b) = abx^{a-1}(1 - x^a)^{b-1}$ and CDF $F(x; a, b) = 1 - (1 - x^a)^b$. With this analytically invertible CDF, one can reparameterize a sample $u$ from the continuous

Uniform$(0,1)$ via the transform $T(u) = (1 - (1-u)^{1/b})^{1/a}$ such that $T(u) \sim$ Kumaraswamy$(a,b)$. Unfortunately, this convenient reparameterization comes at a cost when we derive $p(x_{o_1}, \ldots, x_{o_K}; \alpha)$, which captures the density of the variable returned by algorithm 1. If, in a manner similar to generating a Dirichlet sample from Beta samples, we let $p_i(v; a_i, b_i) \equiv$ Kumaraswamy$(x; \alpha_i, \sum_{j=i+1}^{K} \alpha_j)$, then the resulting variable's density is no longer order agnostic (exchangeable). The exponential in the Kumaraswamy's $(1 - x^a)$ term that admits analytic inverse-CDF sampling, can no longer cancel out $\det(J_{T_i^{-1}})$ terms as the $(1-x)$ term in the Beta analog could. In the simplest case, the 1-simplex ($K = 2$), the possible orderings for algorithm 1 are $o \in O = \{\{1,2\}, \{2,1\}\}$. Indeed, algorithm 1 returns two distinct densities according to their respective orderings:

$$f_{12}(x; a, b) = \alpha_1 \alpha_2 x_1^{\alpha_1 - 1} x_2^{\alpha_2 - 1} \left( \frac{1 - x_1^{\alpha_1}}{1 - x_1} \right)^{\alpha_2 - 1} \tag{3}$$

$$f_{21}(x; a, b) = \alpha_1 \alpha_2 x_1^{\alpha_1 - 1} x_2^{\alpha_2 - 1} \left( \frac{1 - x_2^{\alpha_2}}{1 - x_2} \right)^{\alpha_1 - 1}. \tag{4}$$

In section 7.2 of the appendix, we derive $f_{12}$ and $f_{21}$ as well as the distribution for the 2-simplex, which has orderings $o \in O = \{\{1,2,3\}, \{1,3,2\}, \{2,1,3\}, \{2,3,1\}, \{3,1,2\}, \{3,2,1\}\}$. For $K > 3$, the algebraic book-keeping gets rather involved. We thus rely on algorithm 1 to succinctly represent the complicated densities over the simplex that describe the random variables generated by a stick-breaking process using the Kumaraswamy distribution as the base (stick-breaking) distribution. Our code repository [§] contains a symbolic implementation of algorithm 1 with the Kumaraswamy that programmatically keeps track of the algebra.

## 2.2 The multivariate Kumaraswamy

We posit that a good surrogate for the Dirichlet will exhibit symmetry (exchangeability) properties identical to the Dirichlet it is approximating. If our stick-breaking distribution, $p_i(v; a_i, b_i)$, cannot achieve symmetry for all values $a_i = b_i > 0$, then it is possible that the samples will exhibit bias (fig. 1). If $x \sim$ Beta$(a, b)$, then $(1-x) \sim$ Beta$(b, a)$. It follows then that when $a = b$, $p(x) = p(1-x)$. Unfortunately, Kumaraswamy$(a, b)$ does not admit such symmetry for all $a = b > 0$. However, hope is not lost. From [6, 8], we have lemma 1.

**Lemma 1** *Given a function $f$ of $n$ variables, one can induce symmetry by taking the sum of $f$ over all $n!$ possible permutations of the variables.*

If we define $f_o(x_{o_1}, \ldots, x_{o_K}; \alpha_{o_1}, \ldots, \alpha_{o_K})$ to be the joint density of the $K$-dimensional random variable returned from algorithm 1 with stick-breaking base distribution as $p_i(v; a_i, b_i) \equiv$ Kumaraswamy$(x; \alpha_i, \sum_{j=i+1}^{K} \alpha_j)$ and some ordering $o$, then our proposed distribution for the $(K-1)$-simplex is

$$\text{MV-Kumaraswamy}(x; \alpha) = \mathop{\mathbb{E}}_{o \sim \text{Uniform}(O)} [f_o(x_{o_1}, \ldots, x_{o_K}; \alpha_{o_1}, \ldots, \alpha_{o_K})], \tag{5}$$

where MV-Kumaraswamy stands for Multivariate Kumaraswamy. Here, $O$ is the set of all possible orderings (permutations) of $\{1, \ldots, K\}$. In the context of [8], we create a U-statistic over the variables $x, \alpha$. The expectation in eq. (5) is a summation since we are uniformly sampling $o$ from a discrete set. We therefore can apply lemma 1 to eq. (5) to prove corollary 1.

**Corollary 1** *Let $S \subseteq \{1, \ldots, K\}$ be the set of indices $i$ where for $i \neq j$ we have $\alpha_i = \alpha_j$. Define $A = \{1, \ldots, K\} \setminus S$. Then, $\mathbb{E}_{o \sim Uniform(O)}[f_o(x_{o_1}, \ldots, x_{o_K}; \alpha_{o_1}, \ldots, \alpha_{o_K})]$ is symmetric across barycentric axes $x_a \ \forall \ a \in A$.*

While the factorial growth ($|O| = K!$) for full symmetry is undesirable, we expect approximate symmetry should arise, in expectation, after $O(K)$ samples. Since the problematic bias occurs during the last stick break, each label ideally experiences an ordering where it is not last; this occurs with probability $\frac{K-1}{K}$. Thus, a label is not last, in expectation, after $\frac{K}{K-1}$ draws from Uniform$(O)$.

---

[§] https://github.com/astirn/MV-Kumaraswamy

Therefore, to satisfy this condition for all labels, one needs $\frac{K^2}{K-1} = O(K)$ samples, in expectation. An alternative, which we discuss and demonstrate below in fig. 4, would be to use the $K$ cyclic orderings (e.g. $\{\{1,2,3\}, \{2,3,1\}, \{3,1,2\}\}$ for $K = 3$) to achieve approximate symmetry (exchangeability).

In fig. 2, we provide 1-simplex examples for varying $\alpha$ that demonstrate the effect ordering has on the Kumaraswamy distributions $f_{12}(x;\alpha)$ and $f_{21}(x;\alpha)$ (respectively in eqs. (3) and (4)). In each example, we plot the symmetrized versions arising from our proposed distribution $\mathbb{E}_o[f_o(x;\alpha)]$ (eq. (5)). For reference, we plot the corresponding Dirichlet$(x;\alpha)$, which is equivalent to Beta$(x_1;\alpha_1,\alpha_2)$ for the 1-simplex. Qualitatively, we observe how effectively our proposed distribution resolves the differences between $f_{12}$ and $f_{21}$ and yields a $\mathbb{E}[f_o(x;\alpha)] \approx$ Dirichlet$(x;\alpha)$.

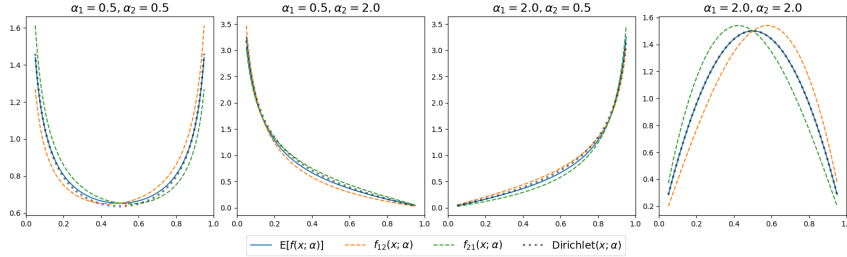

Figure 2: Kumaraswamy asymmetry and symmetrization examples on the 1-simplex.

In fig. 3, we employ Beta distributed stick breaks to generate a Dirichlet random variable. In this example, we pick an $\alpha$ such that the resulting density should be symmetric only about the barycentric $x_1$ axis. Furthermore, because the resulting density is a Dirichlet, the densities arising from all possible orderings should be identical with identical barycentric symmetry properties. The first row contains densities. The subsequent rows measure asymmetry across the specified barycentric axis by computing the absolute difference of the PDF folded along that axis. The first column is for expectation over all possible orderings. The second column is for the expectation over the cyclic orderings. Each column thereafter represents a different stick-breaking order. Indeed, we find that the Dirichlet has an order agnostic density with symmetry only about the barycentric $x_1$ axis.

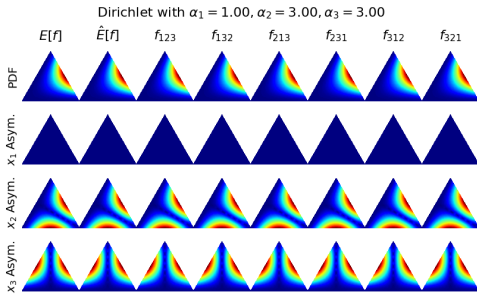

Figure 3: 2-simplex with Beta sticks

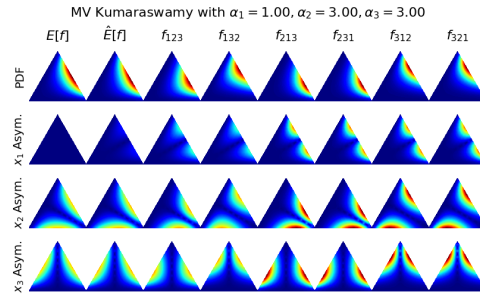

Figure 4: 2-simplex with Kumaraswamy sticks

In fig. 4, we employ the same methodology with the same $\alpha$ as in fig. 3, but this time we use Kumaraswamy distributed stick breaks. Note the significant variations among the densities resulting from the different orderings. It follows that symmetry/asymmetry too vary with respect to ordering. We only see the desired symmetry about the barycentric $x_1$ axis when we take the expectation over all orderings. This example qualitatively illustrates corollary 1. However, we do achieve approximate symmetry when we average over the $K$ cyclic orderings–suggesting we can, in practice, get away with linearly scaling complexity.

## 3 Gradient Variance

We compare our method's gradient variance to other non-explicit gradient reparameterization methods: Implicit Reparameterization Gradients (IRG) [4], RSVI [18], and Generalized Reparameterization

Gradient (GRG) [22]. These works all seek gradient methods with low variance. In fig. 5, we compare MV-Kumaraswamy's (MVK) gradient variance to these other methods by leveraging techniques and code from [18]. Specifically, we consider their test that fits a variational Dirichlet posterior to Categorical data with a Dirichlet prior. In this conjugate setting, true analytic gradients can be computed. Their reported 'gradient variance' is actually the mean square error with respect to the true gradient. In our test, however, we are fitting a MV-Kumaraswamy variational posterior. Therefore, we compute gradient variance, for all methods, according to variance's more common definition. Our tests show that IRG and RSVI ($B = 10$) offer similar variance; this result matches findings in [4].

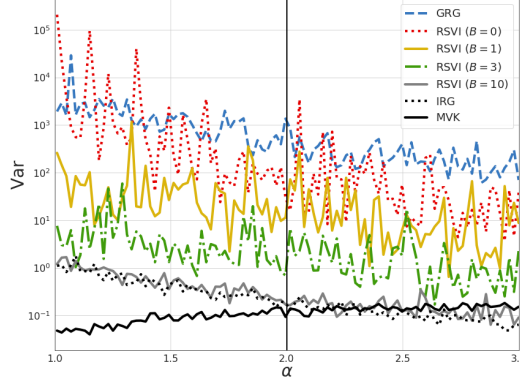

Figure 5: Variance of the ELBO's gradient's first dimension for GRG [22], RSVI [18], IRG [4], and MVK (ours) when fitting a variational posterior to Categorical data with 100 dimensions and a Dirichlet prior. They fit a Dirichlet. We fit a MV-Kumaraswamy using $K = 100$ samples from Uniform($O$) to Monte-Carlo approximate the full expectation; this corresponds to linear complexity.

## 4 A single generative model for semi-supervised learning

We demonstrate the utility of the MV-Kumaraswamy in the context of a parsimonious generative model for semi-supervised learning, with observed data $x$, partially observable classes/labels $y$ with prior $\pi$ and latent variable $z$, all of which are local to each data point. We specify,

$$\pi \sim \text{Dirichlet}(\pi; \alpha), \qquad\qquad\qquad z \sim \mathcal{N}(z; 0, I),$$
$$y|\pi \sim \text{Discrete}(y; \pi), \qquad\qquad\qquad x|y, z \sim p(x|f_\theta(y, z)),$$

where $f_\theta(y, z)$ is a neural network, with parameters $\theta$, operating on the latent variables. For observable $y$, the evidence lower bound (ELBO) for a mean-field posterior approximation $q(\pi, z) = q(\pi)q(z)$ is

$$\ln p(x, y) \geq \mathop{\mathbb{E}}_{q(\pi, z)}[\ln p(x|f_\theta(y, z)) + \ln \pi_y] - D_{KL}(q(\pi) \,||\, p(\pi)) - D_{KL}(q(z) \,||\, p(z))$$
$$\equiv \mathcal{L}_l(x, y, \phi, \theta). \tag{6}$$

For latent $y$, we can derive an alternative ELBO that corresponds to the same generative process of eq. (6), by reintroducing $y$ via marginalization. We derive eqs. (6) and (7) in section 7.3 of the appendix.

$$\ln p(x) \geq \mathop{\mathbb{E}}_{q(\pi, z)}\left[\ln \sum_y p(x|f_\theta(y, z))\pi_y\right] - D_{KL}(q(\pi) \,||\, p(\pi)) - D_{KL}(q(z) \,||\, p(z))$$
$$\equiv \mathcal{L}_u(x, \phi, \theta) \tag{7}$$

Let $L$ be our set of labeled data and $U$ be our unlabeled set. We then consider a combined objective

$$\mathcal{L} = \frac{1}{|L|} \sum_{(x,y) \in L} \mathcal{L}_l(x, y, \phi, \theta) + \frac{1}{|U|} \sum_{x \in U} \mathcal{L}_u(x, \phi, \theta) \tag{8}$$

$$\approx \frac{1}{B} \sum_{(x_i, y_i) \sim L \;\forall\; i \in [B]} \mathcal{L}_l(x_i, y_i, \phi, \theta) + \frac{1}{B} \sum_{x_i \sim U \;\forall\; i \in [B]} \mathcal{L}_u(x_i, \phi, \theta) \tag{9}$$

that balances the two ELBOs evenly. Of concern is when $|U| \gg |L|$. Here, the optimizer could effectively ignore $\mathcal{L}_l(x, y, \phi, \theta)$. This possibility motivates our rebalancing in eq. (8). During optimization we employ batch updates of size $B$ to maximize eq. (9), which similarly balances the contribution between $U$ and $L$. We define an epoch to be the set of batches (sampled without replacement) that constitute $U$. Therefore, when $|U| \gg |L|$, the optimizer will observe samples from $L$ many more times than samples from $U$. Intuitively, the data with observable labels in conjunction with eq. (6) breaks symmetry and encourages the correct assignment of classes to labels.

Following [12, 13], we use an inference network with parameters $\phi$ and define our variational distribution $q(z) = \mathcal{N}(z; \mu_\phi(x), \Sigma_\phi(x))$, where $\mu_\phi(x)$ and $\Sigma_\phi(x)$ are outputs of a neural network operating on the observable data. We restrict $\Sigma_\phi(x)$ to output a diagonal covariance and use a softplus, $\ln(\exp(x) + 1)$, output layer to constrain it to the positive reals. Since $\mu_\phi(x) \in \mathbb{R}^{\dim(z)}$, we use an affine output layer. We let $q(\pi) = \text{MV-Kumaraswamy}(\pi; \alpha_\phi(x))$, where $\alpha_\phi(x)$ is also an output of our inference network. We similarly restrict $\alpha_\phi(x)$ to the positive reals via the softplus activation.

We evaluate the expectations in eqs. (6) and (7) using Monte-Carlo integration. For $q(z)$, we sample from $\mathcal{N}(0, I)$ and utilize the reparameterization trick. Since $q(\pi)$ contains an expectation over orderings, we first sample $o \sim \text{Uniform}(O)$ and then employ algorithm 1 with $p_i(v; a_i, b_i) \equiv \text{Kumaraswamy}(x; \alpha_i, \sum_{j=i+1}^{K} \alpha_j)$, for which we use inverse-CDF sampling. In both cases, gradients are well defined with respect to the variational parameters.

We can decompose $D_{KL}(\text{MV-Kumaraswamy}(\alpha_\phi(x)) \| \text{Dirichlet}(\alpha))$ into a sum over the corresponding Kumaraswamy and Beta stick-breaking distributions as in [20]. Let $\alpha_\phi^{(j)}(x)$ be the $j^{th}$ concentration parameter of the inference network, and $\alpha^{(j)}$ be $j^{th}$ parameter of the Dirichlet prior. If, as above, we let $p(o) = \text{Uniform}(O)$ for the set of all orderings $O$, then $D_{KL}(q(\pi; \alpha_\phi(x)) \| p(\pi; \alpha)) =$

$$\mathbb{E}_{p(o)} \left[ \sum_{i=1}^{K-1} D_{KL}\left( \text{Kumaraswamy}\left( \alpha_\phi^{(o_i)}(x), \sum_{j=i+1}^{K} \alpha_\phi^{(o_j)}(x) \right) \, \middle\| \, \text{Beta}\left( \alpha^{(o_i)}, \sum_{j=i+1}^{K} \alpha^{(o_j)} \right) \right) \right]$$

We compute $D_{KL}(\text{Kumaraswamy}(a, b) \| \text{Beta}(a', b'))$ analytically as in [20] with a Taylor approximation order of 5. We too approximate this expectation with far fewer than $K!$ samples from $p(o)$. Please see section 7.4 of the appendix for a reproduction of this KL-Divergence's mathematical form.

## 5 Experiments

We consider a variety of baselines for our semi-supervised model. Since our work expounds and resolves the order dependence of the original Kumaraswamy stick-breaking construction [20] that uses fixed and constant ordering, we employ their construction (Kumar-SB) as a baseline, for which we force our implementation to use a fixed and constant order during the stick-breaking procedure. As noted in section 1, our model is similar to the M2 model [13]. We too consider it an important baseline for our semi-supervised experiments. Additionally, we use the Softmax-Dirichlet sampling approximation [25]. This approximation forces logits sampled from a Normal variational posterior onto the simplex via the softmax function. In this case, the Dirichlet prior is approximated with a prior for the Gaussian logits [25]. However, this softmax approximation struggles to capture sparsity because the Gaussian prior cannot achieve the multi-modality available to the Dirichlet [22]. Lastly, we include a comparison to Implicit Reparameterization Gradients (IRG) [4]. Here, we set $q(\pi; \alpha_\phi(x)) = \text{Dirichlet}(\pi; \alpha_\phi(x))$ in our semi-supervised model with the same architecture. IRG uses independent Gamma samples to construct Beta and Dirichlet samples. IRG's principle contribution for gradient reparameterization is that it side-steps the need to invert the standardization function (i.e. the CDF). However, IRG still requires Gamma CDF gradients w.r.t. the variational parameters. These gradients do not have a known analytic form, mandating their application of forward-mode automatic differentiation to a numerical method. In our IRG baseline, both the prior and variational posterior are Dirichlet distributions yielding an analytic KL-Divergence. We mention but do not test [9], which similarly constructs Dirichlet samples from normalized Gamma samples. They too employ implicit differentiation to avoid differentiating the inverse CDF, but necessarily fall back to numerically differentiating the Gamma CDF.

Our source code can be found at `https://github.com/astirn/MV-Kumaraswamy`. For our latest experimental results, please refer to `https://arxiv.org/abs/1905.12052`. In our generative

process and eqs. (6) and (7), we referred generally to our data likelihood as $p(x|f_\theta(y,z))$. In all of our experiments, we assume $p(x|f_\theta(y,z)) = \mathcal{N}(x, \mu_\theta(y,z), \Sigma_\theta(y,z))$, where $\mu_\theta(y,z)$ and $\Sigma_\theta(y,z)$ are outputs of a neural network with parameters $\theta$ operating on the latent variables. We use diagonal covariance for $\Sigma_\theta(y,z)$. Across all of our experiments, we maintain consistent recognition and generative network architectures, which we detail in section 7.5 of the appendix.

We do not use any explicit regularization. Our models are implemented in TensorFlow and were trained using ADAM with a batch size $B = 250$ and 5 Monte-Carlo samples for each training example. We use learning rates $1 \times 10^{-3}$ and $1 \times 10^{-4}$ respectively for MNIST and SVHN. Other optimizer parameters were kept at TensorFlow defaults. We utilized GPU acceleration and found that cards with $\sim$8 GB of memory were sufficient. We utilize the TensorFlow Datasets API, from which we source our data. For all experiments, we split our data into 4 subsets: unlabeled training ($U$) data, labeled training ($L$) data, validation data, and test data. For MNIST: $|U| = 49,400$, $|L| = 600$, $|\text{validation}| = |\text{test}| = 10,0000$. For SVHN: $|U| = 62,257$, $|L| = 1000$, $|\text{validation}| = 10,000$, $|\text{test}| = 26,032$. When constructing $L$, we enforce label balancing. We allow all trials to train for a maximum of 750 epochs, but use validation set performance to enable early stopping whenever the loss (eq. (8)) and classification error have not improved in the previous 15 epochs. All reported metrics were collected from the test set during the validation set's best epoch–we do this independently for classification error and log likelihood. For each trial, all models utilize the same random data split except where noted[†]. We translate the uint8 encoded pixel intensities to $[0,1]$ by dividing by 255, but perform no other preprocessing.

Table 1: Held-out test set classification errors and log likelihoods. A "$--$" for a $p$-value indicates it was unavailable either because it was with respect to itself or the corresponding data and/or number of trials were missing. Since [13] did not report log likelihoods, we did not collect them with our implementation.

| Experiment | Method | Error | $p$-value | Log Likelihood | $p$-value |
|---|---|---|---|---|---|
| MNIST | MV-Kum. | $0.099 \pm 0.011$ | $--$ | $-6.4 \pm 6.3$ | $--$ |
| 10 trials | IRG[4] | $0.097 \pm 0.008$ | $0.72$ | $-7.8 \pm 7.1$ | $0.64$ |
| 600 labels | Kumar-SB[20] | $0.248 \pm 0.009$ | $1.05 \times 10^{-17}$ | $-6.5 \pm 6.3$ | $0.95$ |
| $\dim(z) = 0$ | Softmax | $0.093 \pm 0.009$ | $0.24$ | $-6.5 \pm 6.2$ | $0.95$ |
| MNIST | MV-Kum. | $0.043 \pm 0.005$ | $--$ | $45.06 \pm 0.92$ | $--$ |
| 10 trials | IRG[4] | $0.044 \pm 0.006$ | $0.89$ | $45.69 \pm 0.38$ | $0.06$ |
| 600 labels | M2 (ours) | $0.098 \pm 0.014$ | $5.37 \times 10^{-10}$ | Not collected | $--$ |
| $\dim(z) = 2$ | Kumar-SB[20] | $0.138 \pm 0.015$ | $1.65 \times 10^{-13}$ | $44.33 \pm 1.65$ | $0.24$ |
| | Softmax | $0.042 \pm 0.003$ | $0.40$ | $45.14 \pm 0.73$ | $0.82$ |
| MNIST | MV-Kum. | $0.018 \pm 0.004$ | $--$ | $116.58 \pm 0.68$ | $--$ |
| 10 trials | IRG[4] | $0.018 \pm 0.004$ | $0.98$ | $116.57 \pm 0.43$ | $0.97$ |
| 600 labels | M2 (ours) | $0.020 \pm 0.003$ | $0.32$ | Not collected | $--$ |
| $\dim(z) = 50$ | Kumar-SB[20] | $0.071 \pm 0.008$ | $2.58 \times 10^{-13}$ | $116.22 \pm 0.33$ | $0.15$ |
| | Softmax | $0.018 \pm 0.003$ | $0.87$ | $116.24 \pm 0.45$ | $0.21$ |
| | M2[†][13] | $0.049 \pm 0.001$ | $--$ | Not reported | $--$ |
| | M1 + M2[†][13] | $0.026 \pm 0.005$ | $--$ | Not reported | $--$ |
| SVHN | MV-Kum. | $0.296 \pm 0.014$ | $--$ | $669.37 \pm 0.57$ | $--$ |
| 4 trials | IRG[4] | $0.288 \pm 0.008$ | $0.38$ | $669.84 \pm 0.84$ | $0.39$ |
| 1000 labels | M2 (ours) | $0.406 \pm 0.027$ | $3.64 \times 10^{-04}$ | Not collected | $--$ |
| $\dim(z) = 50$ | Kumar-SB[20] | $0.702 \pm 0.011$ | $7.42 \times 10^{-09}$ | $669.44 \pm 0.77$ | $0.89$ |
| | Softmax | $0.300 \pm 0.007$ | $0.61$ | $669.51 \pm 0.72$ | $0.78$ |
| | M1 + M2[†][13] | $0.360 \pm 0.001$ | $--$ | Not reported | $--$ |

For the semi-supervised learning task, we present classification and reconstruction performances in table 1 using our algorithm as well as the baselines discussed previously. We organize our results by experiment group. All reported $p$-values are with respect to our MV-Kumaraswamy model's performance for corresponding $\dim(z)$. We say, "M2 (ours)," whenever we use the generative process of [13] with our neural network architecture. For a subset of experiments, we present results from [13]–without knowing how many trials they ran we cannot compute the corresponding $p$-value. We recognize that there are numerous works [21, 1, 26, 15, 10, 7, 24, 2, 16, 17] that offer superior

performance on these tasks, however, we abstain from reporting these performances whenever those models are not variational Bayesian, use adversarial training, lack explicit generative processes, use architectures vastly larger in size than ours, or use a different number of labeled examples ($\neq 600$ for MNIST and $\neq 1000$ for SVHN).

In fig. 6, we plot the latent space representation for our MV-Kumaraswamy model for MNIST when $\dim(z) = 2$. Each digit's manifold is over $(-1.5, -1.5) \times (1.5, 1.5)$, which corresponds to $\pm 1.5$ standard deviations from the prior. The only difference in latent encoding between corresponding manifold positions is the label provided to the generative network. Interestingly, the model learns to use $z$ in a qualitatively similar way to represent character transformations across classes.

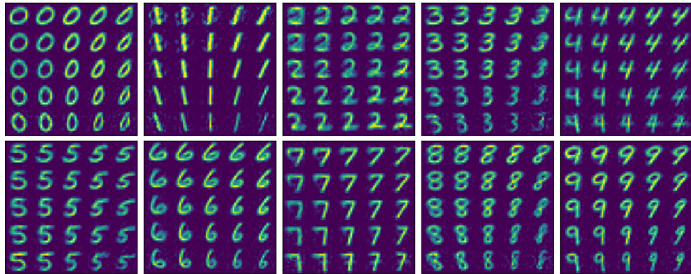

Figure 6: Latent space for MV-Kumaraswamy model with $\dim(z) = 2$.

## 6    Discussion

The statistically significant classification performance gains of MV-Kumaraswamy (approximate integration over all orderings) against Kumar-SB [20] (fixed and constant ordering) validates the impact of our contribution. Kumar-SB's worse performance is likely due to the over allocation of probability mass to the final stick during sampling (fig. 1). When the class-assignment posterior has high entropy, the fixed order sampling will bias the last label dimension. Further, MV-Kumaraswamy beats [13] for both classification tasks despite our single model approach and minimal preprocessing. Interestingly, our implementation of M2 seemingly requires a larger $\dim(z)$ to match the classification performance of MV-Kumaraswamy. Lastly, IRG's classification performance is not statistically distinguishable from ours. Deep learning frameworks' (e.g. TensorFlow, PyTorch, Theano, CNTK, MXNET, Chainer) distinct advantage is NOT requiring user-computed gradients. We argue that methods requiring numerical gradients [4, 9] do not admit a straightforward implementation for the common practitioner as they require additional (often non-trivial) code to supply the gradient estimates to the framework's optimizer. Conversely, our method has analytic gradients, enabling easy integration into ANY deep learning framework. To the best of our knowledge, IRG for the Gamma, Beta, and Dirichlet distributions only exists in TensorFlow (IRG was developed at Deep Mind).

VAEs offer scalable and efficient learning for a subset of Bayesian models. Applied Bayesian modeling, however, makes heavy use of distributions outside this subset. In particular, the Dirichlet, without some form of accommodation or approximation, will render a VAE intractable since gradients with respect to variational parameters are challenging to compute. Efficient approximation of such gradients is an active area of research. However, explicit reparameterization is advantageous in terms of simplicity and efficiency. In this article, we present and develop theory for a computationally efficient and explicitly reparameterizable Dirichlet surrogate that has similar sparsity-inducing capabilities and identical exchangeability properties to the Dirichlet it is approximating. We confirm its surrogate candidacy through a range of semi-supervised auto-encoding tasks. We look forward to utilizing our new distribution to scale inference in more structured probabilistic models such as topic models. We hope others will use our distribution not only as a surrogate for a Dirichlet posterior but also as a prior. The latter might yield a more exact divergence between the variational posterior and its prior.

**Acknowledgments**

This work was supported in part by NSF grant III-1526914.

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
