[Supplementary Material]

# 7 Appendix

## 7.1 Stick-Breaking: Beta to Dirichlet

In this section, we prove theorem 1. Prior to executing the **while** loop, algorithm 1 samples $v_{o_1} \sim$ Beta $\left(\alpha_{o_1}, \sum_{j=2}^{K} \alpha_{o_j}\right)$ and assigns $x_{o_1} \leftarrow v_{o_1}$. Therefore, $x_{o_1}$ has density

$$p(x_{o_1}) = \frac{\Gamma\left(\sum_{j=1}^{K} \alpha_{o_j}\right)}{\Gamma(\alpha_{o_1})\Gamma\left(\sum_{j=2}^{K} \alpha_{o_j}\right)} x_{o_1}^{\alpha_{o_1}-1}(1-x_{o_1})^{\left(\sum_{j=2}^{K} \alpha_{o_j}\right)-1}. \tag{10}$$

In the case $K = 2$, algorithm 1 does not execute the **while** loop and concludes after assigning $x_{o_2} \leftarrow 1 - x_{o_1}$. From one perspective, algorithm 1 returns a 2-dimensional variable whose density is fully determined by the first dimension (the only degree of freedom for the 1-simplex). In the $K = 2$ case, this univariate density is the only utilized base distribution, the Beta$(x; \alpha_{o_1}, \alpha_{o_2})$. However, if one wants to incorporate $x_{o_2}$ into this density, one can substitute $x_{o_2}$ for $1 - x_{o_1}$ as follows:

$$p(x_{o_1}, x_{o_2}) = \frac{\Gamma(\alpha_{o_1} + \alpha_{o_2})}{\Gamma(\alpha_{o_1})\Gamma(\alpha_{o_2})} x_{o_1}^{\alpha_{o_1}-1} x_{o_2}^{\alpha_{o_2}-1} = \text{Dirichlet}(x; \alpha).$$

Thus, we have proved correctness of algorithm 1 for $K = 2$. For $K > 2$, algorithm 1 will execute the **while** loop. Therefore, we will use induction to prove loop correctness. At the $i^{th}$ iteration of the loop, algorithm 1 samples $v_{o_i} \sim$ Beta $\left(\alpha_{o_i}, \sum_{j=i+1}^{K} \alpha_{o_j}\right)$ and assigns $x_{o_i} \leftarrow v_{o_i}\left(1 - \sum_{j=1}^{i-1} x_{o_j}\right)$. Using eq. (2) as the inverse to our change-of-variables transformation, we can claim, at the $i^{th}$ iteration of the loop, that $p(x_{o_i}|x_{o_{i-1}}, \ldots, x_{o_1})$

$$= \text{Beta}\left(x_{o_i}\left(1 - \sum_{j=1}^{i-1} x_{o_j}\right)^{-1}; \alpha_{o_i}, \sum_{j=i+1}^{K} \alpha_{o_j}\right)\left(1 - \sum_{j=1}^{i-1} x_{o_j}\right)^{-1}$$

$$= \frac{\Gamma\left(\sum_{j=i}^{K} \alpha_{o_j}\right)}{\Gamma(\alpha_{o_i})\Gamma\left(\sum_{j=i+1}^{K} \alpha_{o_j}\right)} \frac{x_{o_i}^{\alpha_{o_i}-1}}{\left(1 - \sum_{j=1}^{i-1} x_{o_j}\right)^{\alpha_{o_i}}} \frac{\left(1 - \sum_{j=1}^{i} x_{o_j}\right)^{\left(\sum_{j=i+1}^{K} \alpha_{o_j}\right)-1}}{\left(1 - \sum_{j=1}^{i-1} x_{o_j}\right)^{\left(\sum_{j=i+1}^{K} \alpha_{o_j}\right)-1}}$$

$$= \frac{\Gamma\left(\sum_{j=i}^{K} \alpha_{o_j}\right)}{\Gamma(\alpha_{o_i})\Gamma\left(\sum_{j=i+1}^{K} \alpha_{o_j}\right)} x_{o_i}^{\alpha_{o_i}-1}\left(1 - \sum_{j=1}^{i-1} x_{o_j}\right)^{1-\left(\sum_{j=i}^{K} \alpha_{o_j}\right)}\left(1 - \sum_{j=1}^{i} x_{o_j}\right)^{\left(\sum_{j=i+1}^{K} \alpha_{o_j}\right)-1}. \tag{11}$$

For $K > 2$, consider the base case where $i = 2$, corresponding to the first of $(K - 2)$ **while** loop iterations. Leveraging eq. (11) for for this initial iteration ($i = 2$), we find that $p(x_{o_2}|x_{o_1})$

$$= \frac{\Gamma\left(\sum_{j=2}^{K} \alpha_{o_j}\right)}{\Gamma(\alpha_{o_2})\Gamma\left(\sum_{j=3}^{K} \alpha_{o_j}\right)} x_{o_2}^{\alpha_{o_2}-1}(1-x_{o_1})^{1-\left(\sum_{j=2}^{K} \alpha_{o_j}\right)}(1 - x_{o_1} - x_{o_2})^{\left(\sum_{j=3}^{K} \alpha_{o_j}\right)-1}.$$

For $K > 2$ and the $i = 2$ base case, multiplying this $p(x_{o_2}|x_{o_1})$ by $p(x_{o_1})$ (eq. (10)) to construct a joint density yields:

$$p(x_{o_1}, x_{o_2}) = p(x_{o_1})p(x_{o_2}|x_{o_1})$$

$$= \frac{\Gamma\left(\sum_{j=1}^{K} \alpha_{o_j}\right)}{\Gamma(\alpha_{o_1})\Gamma(\alpha_{o_2})\Gamma\left(\sum_{j=3}^{K} \alpha_{o_j}\right)} x_{o_1}^{\alpha_{o_1}-1} x_{o_2}^{\alpha_{o_2}-1}(1 - x_{o_1} - x_{o_2})^{\left(\sum_{j=3}^{K} \alpha_{o_j}\right)-1}.$$

Indeed, $p(x_{o_1}, x_{o_2})$ can be viewed as Dirichlet$(x_{o_1}, x_{o_2}, x_{o_3}; \alpha_{o_1}, \alpha_{o_2}, \sum_{j=3}^{K} \alpha_{o_j})$ after substituting $x_{o_3}$ for $1 - x_{o_1} - x_{o_2}$. Recall that we already proved $p(x_{o_1})$ is Dirichlet (eq. (10)). Consequently, the **while** loop is guaranteed to begin with a Dirichlet. Just now, we proved the joint density after the

first **while** loop iteration also is Dirichlet. Because algorithm 1 concludes **while** loop execution after $i = K - 1$, if we can prove for subsequent iterations (the inductive step) that the density is also a Dirichlet, then we have completed the proof via induction. Similar to the $i = 2$ base case, one can write the joint density resulting after the $i^{th}$ loop iteration as $p(x_{o_1}, \ldots, x_{o_i})$

$$= \frac{\Gamma\left(\sum_{j=1}^{K} \alpha_{o_j}\right)}{\left(\prod_{j=1}^{i} \Gamma(\alpha_{o_i})\right)\Gamma\left(\sum_{j=i+1}^{K} \alpha_{o_j}\right)} \left(\prod_{j=1}^{i} x_{o_i}^{\alpha_{o_i}-1}\right)\left(1 - \sum_{j=1}^{i} x_{o_j}\right)^{\left(\sum_{j=i+1}^{K} \alpha_{o_j}\right)-1}.$$

The next **while** loop iteration has conditional density $p(x_{o_{i+1}}|x_{o_i}, \ldots, x_{o_1})$, which when multiplied by $p(x_{o_1}, \ldots, x_{o_i})$, yields a joint density $p(x_{o_1}, \ldots, x_{o_{i+1}})$

$$= \frac{\Gamma\left(\sum_{j=1}^{K} \alpha_{o_j}\right)}{\left(\prod_{j=1}^{i+1} \Gamma(\alpha_{o_i})\right)\Gamma\left(\sum_{j=i+2}^{K} \alpha_{o_j}\right)} \left(\prod_{j=1}^{i+1} x_{o_i}^{\alpha_{o_i}-1}\right)\left(1 - \sum_{j=1}^{i+1} x_{o_j}\right)^{\left(\sum_{j=i+2}^{K} \alpha_{o_j}\right)-1}.$$

Substituting $x_{o_{i+2}}$ for $1 - \sum_{j=1}^{i+1} x_{o_j}$, yields Dirichlet$(x_{o_1}, \ldots, x_{o_{i+2}}; \alpha_{o_1}, \ldots, \alpha_{o_{i+1}}, \sum_{j=i+2}^{K} \alpha_{o_j})$. Hence, we have completed a proof of theorem 1.

## 7.2 Stick-Breaking: Kumaraswamy

In this section, we derive, in the cases of the 1-simplex and the 2-simplex, the density of the random variable return by algorithm 1 when $p_i(v; a_i, b_i) \equiv$ Kumaraswamy $\left(x; \alpha_i, \sum_{j=i+1}^{K} \alpha_j\right)$.

### 7.2.1 Stick-Breaking: Kumaraswamy to a 1-Simplex Distribution

In the case $K = 2$, algorithm 1 begins by sampling $v_{o_1} \sim$ Kumaraswamy$(\alpha_{o_1}, \alpha_{o_2})$ and assigning $x_{o_1} \leftarrow v_{o_1}$. Therefore, $x_{o_1}$ has density

$$p(x_{o_1}) = \alpha_{o_1}\alpha_{o_2}x_{o_1}^{\alpha_{o_1}-1}\left(1 - x_{o_1}^{\alpha_{o_1}}\right)^{\alpha_{o_2}-1}.$$

Because $K = 2$, algorithm 1 does not execute the **while** loop and concludes by assigning $x_{o_2} \leftarrow 1 - x_{o_1}$. From one perspective, algorithm 1 returns a 2-dimensional variable whose density is fully determined by the first dimension (the only degree of freedom for the 1-simplex). In the $K = 2$ case, this univariate density is the only utilized base distribution, the Kumaraswamy$(x; \alpha_{o_1}, \alpha_{o_2})$. However, if one wants to incorporate $x_{o_2}$ into the density, one can do so by multiplying by 1 as follows:

$$p(x_{o_1}, x_{o_2}) = p(x_{o_1})\left(\frac{x_{o_2}}{1 - x_{o_1}}\right)^{\alpha_{o_2}-1}$$

$$= \alpha_{o_1}\alpha_{o_2}x_{o_1}^{\alpha_{o_1}-1}x_{o_2}^{\alpha_{o_2}-1}\left(\frac{1 - x_{o_1}^{\alpha_{o_1}}}{1 - x_{o_1}}\right)^{\alpha_{o_2}-1}.$$

As mentioned in section 2.1, the $(1 - x^a)$ term in the Kumaraswamy distribution induces algebraic complexities that do not cancel out (in opposition to the case of the Beta distribution).

### 7.2.2 Stick-Breaking: Kumaraswamy to a 2-Simplex Distribution

In the case $K = 3$, algorithm 1 begins by sampling $v_{o_1} \sim$ Kumaraswamy$(\alpha_{o_1}, \alpha_{o_2} + \alpha_{o_3})$ and assigning $x_{o_1} \leftarrow v_{o_1}$. Therefore, $x_{o_1}$ has density

$$p(x_{o_1}) = \alpha_{o_1}(\alpha_{o_2} + \alpha_{o_3})x_{o_1}^{\alpha_{o_1}-1}\left(1 - x_{o_1}^{\alpha_{o_1}}\right)^{\alpha_{o_2}+\alpha_{o_3}-1}.$$

Thereafter, algorithm 1 enters the **while** loop at $i = 2$ and samples $v_{o_2} \sim$ Kumaraswamy$(\alpha_{o_2}, \alpha_{o_3})$ and assigns $x_{o_2} \leftarrow v_{o_2}(1 - x_{o_1})$. Using eq. (2) as the inverse to our change-of-variables transforma-

tion, we can claim

$$p(x_{o_2}|x_{o_1}) = \text{Kumaraswamy}\left(\frac{x_{o_2}}{1-x_{o_1}}; \alpha_{o_2}, \alpha_{o_3}\right)(1-x_{o_1})^{-1}$$

$$= \alpha_{o_2}\alpha_{o_3}\left(\frac{x_{o_2}}{1-x_{o_1}}\right)^{\alpha_{o_2}-1}\left(1-\left(\frac{x_{o_2}}{1-x_{o_1}}\right)^{\alpha_{o_2}}\right)^{\alpha_{o_3}-1}(1-x_{o_1})^{-1}$$

$$= \alpha_{o_2}\alpha_{o_3}x_{o_2}^{\alpha_{o_2}-1}(1-x_{o_1})^{-\alpha_{o_2}}\left(1-\left(\frac{x_{o_2}}{1-x_{o_1}}\right)^{\alpha_{o_2}}\right)^{\alpha_{o_3}-1}.$$

With $K = 3$, the **while** loop only performs a single iteration. With all iterations complete, we can construct the joint distribution as follows:

$$p(x_{o_1}, x_{o_2}) = p(x_{o_1})p(x_{o_2}|x_{o_1})$$

$$= \left[\prod_{i=1}^{3}\alpha_{o_i}\right](\alpha_{o_2}+\alpha_{o_3})\left[\prod_{i=1}^{2}x_{o_i}^{\alpha_{o_i}-1}\right]\frac{(1-x_{o_1})^{-\alpha_{o_2}}\left(1-x_{o_1}^{\alpha_{o_1}}\right)^{\alpha_{o_2}+\alpha_{o_3}-1}}{\left(1-\left(\frac{x_{o_2}}{1-x_{o_1}}\right)^{\alpha_{o_2}}\right)^{1-\alpha_{o_3}}}.$$

Unfortunately, this joint density does not admit an easy substitution for algorithm 1's final step of assigning $x_{o_3} \leftarrow 1 - x_{o_1} - x_{o_2}$. We therefore leave $p(x_{o_1}, x_{o_2}, x_{o_3})$ as a function of just $x_{o_1}$ and $x_{o_2}$ and in the form of $p(x_{o_1}, x_{o_2})$, which is consistent with the fact that $x_{o_3}$ is deterministic given $x_{o_1}$ and $x_{o_2}$. In other words, $x_{o_1}$ and $x_{o_2}$ are the 2 degrees of freedom for the 2-simplex.

### 7.3 Model Derivation

In this section, we derive the evidence lower bound (ELBO) for the generative process outlined in the beginning of section 4 and the corresponding mean-field posterior approximation $q(\pi, z) = q(\pi)q(z)$. In the case of observable $y$, we find that

$$\ln p(x, y) = \ln p(x|y, z) + \ln p(y|\pi) + \ln p(\pi) + \ln p(z) - \ln p(\pi, z|x, y)$$

$$= \ln p(x|y, z) + \ln p(y|\pi) - \ln\frac{q(\pi)}{p(\pi)} - \ln\frac{q(z)}{p(z)} + \ln\frac{q(\pi, z)}{p(\pi, z|x, y)}$$

$$= \mathop{\mathbb{E}}_{q(\pi,z)}[\ln p(x|f_\theta(y, z)) + \ln\pi_y] - D_{KL}(q(\pi) \,||\, p(\pi))$$

$$\quad - D_{KL}(q(z) \,||\, p(z)) + D_{KL}(q(\pi, z) \,||\, p(\pi, z|x, y))$$

$$\geq \mathop{\mathbb{E}}_{q(\pi,z)}[\ln p(x|f_\theta(y, z)) + \ln\pi_y] - D_{KL}(q(\pi) \,||\, p(\pi)) - D_{KL}(q(z) \,||\, p(z))$$

$$\equiv \mathcal{L}_l(x, y, \phi, \theta).$$

In the case that $y$ is latent, we can derive an alternative ELBO with the same mean-field posterior approximation as above:

$$
\begin{aligned}
\ln p(x) &= \ln p(x|\pi, z) + \ln p(\pi) + \ln p(z) - \ln p(\pi, z|x) \\
&= \ln p(x|\pi, z) - \ln \frac{q(\pi)}{p(\pi)} - \ln \frac{q(z)}{p(z)} + \ln \frac{q(\pi, z)}{p(\pi, z|x)} \\
&= \mathop{\mathbb{E}}_{q(\pi, z)}[\ln p(x|\pi, z)] - D_{KL}(q(\pi) \,||\, p(\pi)) \\
&\quad - D_{KL}(q(z) \,||\, p(z)) + D_{KL}(q(\pi, z) \,||\, p(\pi, z|x)) \\
&\geq \mathop{\mathbb{E}}_{q(\pi, z)}[\ln p(x|\pi, z)] - D_{KL}(q(\pi) \,||\, p(\pi)) - D_{KL}(q(z) \,||\, p(z)) \\
&= \mathop{\mathbb{E}}_{q(\pi, z)}\left[ \ln \sum_y p(x, y|\pi, z) \right] - D_{KL}(q(\pi) \,||\, p(\pi)) - D_{KL}(q(z) \,||\, p(z)) \\
&= \mathop{\mathbb{E}}_{q(\pi, z)}\left[ \ln \sum_y p(x|y, z)p(y|\pi) \right] - D_{KL}(q(\pi) \,||\, p(\pi)) - D_{KL}(q(z) \,||\, p(z)) \\
&= \mathop{\mathbb{E}}_{q(\pi, z)}\left[ \ln \sum_y p(x|f_\theta(y, z))\pi_y \right] - D_{KL}(q(\pi) \,||\, p(\pi)) - D_{KL}(q(z) \,||\, p(z)) \\
&\equiv \mathcal{L}_u(x, \phi, \theta).
\end{aligned}
$$

### 7.4  Kumaraswamy-Beta KL-Divergence

For convenience, we reproduce (from [20]) the KL-Divergence between the Kumarswamy and Beta distributions. In particular, $D_{KL}(\text{Kumaraswamy}(a, b) \,||\, \text{Beta}(\alpha, \beta)) =$

$$
\frac{a - \alpha}{a}\left( -\gamma - \Psi(b) - \frac{1}{b} \right) + \log(ab) + \log B(\alpha, \beta) - \frac{b-1}{b} + (\beta - 1)b \sum_{m=1}^{\infty} \frac{1}{m + ab} B\left( \frac{m}{a}, b \right)
$$

where $\gamma$ is Euler's constant, $\Psi(\cdot)$ is the Digamma function, and $B(\cdot, \cdot)$ is the Beta function. An $n$'th order Taylor approximation of the above occurs when one replaces the infinite summation with the summation over the first $n$ terms.

### 7.5  Network Architecture

Our experiments exclusively utilize image data. For convenience, we define $w_x$ and $l_x$ as the width and length of the input images. Our inference network's hidden layers are:

1. Convolution layer with a $5 \times 5 \times (5 \cdot \text{number of data channels})$ kernel followed by an exponential linear (ELU) [3] activation and a $3 \times 3$ max pool with a stride of 2
2. Convolution layer with a $3 \times 3 \times (10 \cdot \text{number of data channels})$ kernel followed by an ELU activation and a $3 \times 3$ max pool with a stride of 2
3-4. A fully-connected layer with 200 outputs with an ELU activation

The last hidden layer serves as input to the output layer, which produces values for $\alpha_\phi(x)$, $\mu_\phi(x)$ and $\Sigma_\phi(x)$ with an affine operation and application of the activations described in section 4. Our generative network's hidden layers are:

1-2. A fully-connected layer with 200 outputs with an ELU activation
3. A fully-connected layer with ELU activations and a reshape to achieve an output of shape $\frac{w_x}{4} \times \frac{l_x}{4} \times (10 \cdot \text{number of data channels})$.
2×4. Convolution transpose layer with a $3 \times 3 \times (5 \cdot \text{number of data channels})$ kernel followed by an ELU activation and a $2\times$ bi-linear up-sample–there are 2 of these layers in parallel, one each for $\mu_\theta(y, z)$ and $\Sigma_\theta(y, z)$.
2×5. Convolution transpose layer with a $5 \times 5 \times (\text{number of data channels})$ kernel followed by an ELU activation and a $2\times$ bi-linear up-sample–as before there are 2 of these layers in parallel

These parameter sizes guarantee that $\mu_\theta(y, z)$ and $\Sigma_\theta(y, z)$ have the same dimensions as $x$. Our model offers the same attractive computational complexities as the original M2 model [13].