[Reviews · NeurIPS 2019]

Reviewer 1



Originality: Although VAEs using a stick-breaking construction with Kumaraswamy distributions has been considered before (Nalisnick, Smyth, STICK-BREAKING VARIATIONAL AUTOENCODERS, 2017), the idea to use such a construction and extend it by mixing over the orderings to obtain a density more similar to a Dirichlet is new and interesting. Related work is adequately cited. Quality: The paper seems technically sound and claims are largely supported. Although Theorem 1 is a standard result, reiterating it is likely useful for the subsequent exposition. Experimental results show that the method outperforms some baselines, however, I feel that some additional experiments would be useful (see details below in Section 5. Improvements). Clarity: The paper is written relatively clearly, with some rather minor issues, see below. Significance: The idea of mixing over the orderings in the stick-breaking process construction can be a quite useful idea - for practitioners it could simplify to computing the gradients using reparameterisation plus some Monte Carlo averaging - although it is not so clear how well it compares to some recent work. Also it could be used for applications different from the VAE considered such as in non-amortized variational inference in Bayesian models with a Dirichlet prior. Some issues: -Can you explain why equation 181 holds and how you compute the KL? There is some dependence over the components when doing the stick-breaking, plus some additional mixing over the orderings, so this is not so obvious to me how this works. -How many Monte Carlo samples over the ordering does one need to get approximately symmetric distributions in practice? Like with a 50 dimensional latent space you are considering, does this not increase the variance of the gradients too much? -Apart from symmetry issues, can there anything be said about what does randomizing over the ordering imply for the moments of the distribution (compared to say y \sim Dirichlet(\alpha) having a negative covariance Cov(y_i,y_j)=-\alpha_i \alpha_j / (\sum_k\alpha_k)^2(\sum_k\alpha_k+1)? Some minor issues: -it might be more clearer to make explicit the conditioning on x in q(\pi) and q(z) in equation 7 -There seems to be some confusing with the letter \pi and x in lines 173 and 181. -After lines 385 and 387, I find it more clearer if the integration over the variational distribution is done earlier than after the third line. #POST AUTHOR RESPONSE: The response of the authors makes the paper stronger, as they include additional experiments comparing the proposed approach with recent alternatives (Figurnov et al., Implicit Reparameterization Gradients, 2018; and Naesseth et al., Reparameterization Gradients through Acceptance-RejectionSampling Algorithms), that I had liked to see as an improvement, so I increase my score from 5 to 6. The proposed method performs well. A comparison with simple score/reinforce-gradient estimator would also be nice as it might show that low-variance gradients are necessary for this specific application. However, the calculation of the KL divergence in line 181 is still not clear to me, even after the response. Particularly, it is not obvious to me why the dependence of x_{o_i} on x_{o_1},…,x_{o_{i-1}} seems to not matter in the KL-calculation, or is this meant be just an approximation?

Reviewer 2



## A New distribution on the Simplex with Auto-Encoding Applications ## Review after author rebuttal The authors significantly strengthen the paper by addressing the comments of the reviewers. In particular, they extended the experimental section (which I was particularly concerned about) by adding all the baselines that I suggested. IRG is the most competitive baseline which seems to have the same performance. The authors argue that the implementation of the proposed method is simpler that IRG. On the other hand IRG is more general (it is not only about the dirichlet distribution). Based on this, I am increasing my score to 6. ## Summary The authors propose a new distribution over the simplex amenable to the re-parameterization trick. To so so, they resort stick-breaking construction sampling the sticks from i.i.d. kumaraswamy distributions. To avoid the influence of the order in which the stick are sampled, they propose to integrate over all possible orders and they resort to MC to estimate this integral. In the experiments, they apply the proposed distribution to approximate the posterior over the labels of a semi-supervised conditional VAE. As baselines they use the original proposal that does not use a prior and using a gaussian-softmax prior. However, the results are incomplete and they fail to compare to other more recent proposals (Gamma-SB-VAE, Kumar-SB-VAE, general reparameterization trick, ...). ### Details The main idea of the authors is to used an stick-breaking construction sampling the sticks from i.i.d. kumaraswamy distributions (and idea already published a few years ago) and get rid off the influence of the order in which the sticks are sampled by integrating over all possible orders. This integration is intractable, so they approximate it using plain montecarlo estimates. This has been applied in other contexts before but I believe the application to symmetrize the kumaraswamy-stick-breaking distribution has not done before to the extend of my knowledge. However, it seems somehow straightforward. The results are show in the first part of the paper, however, even thought it is technically correct it seems to me that claimming a "new distribution" it is maybe an oversell. Nevertheless, the main weakness of the paper is when the try to prove the superiority of this distribution when applied as a prior of a model. The authors choose the semi-supervised conditional autoencoder originally proposed in [1]. In the original proposal, Kingma et al do not use a prior over the conditioning categorical variables. In this paper, the authors propose to use the symetrized-kumaraswamy-stick-breaking and they show that it improves the performance. The also compare to a gaussian-softmax prior for which it is know that cannot model multi-modal distributions and again the symetrized-kumaraswamy-stick-breaking slightly outperforms this prior. However, the authors do not compare to the most obvious baseline, the kumaraswamy-stick-breaking already proposed in [2]. This baseline is needed to see the actual contribution of the paper which is the approximate integration over all possible orders, not the kumaraswamy-stick-breaking contruction that have been already used in several papers in the literature. Also, in [2] they use a gamma-stick-breaking construction based on an approximation of the inverse CDF. Finally, there have been some interesting advances that extends the re-parameterization trick to the beta/gamma distribution [3, 4] that are also missing in the experimental section. Overall, the theoretical contribution is not novel enough and the experimental section is far from complete. It could be a candidate for a workshop paper but it falls below the novelty and quality neurips bar. ### Minors * Why the results table seems incomplete? * Why using a beta in the KL term when you could used the symetrized-kumaraswamy-stick-breaking as well? ### References [1] Durk P Kingma, Shakir Mohamed, Danilo Jimenez Rezende, and Max Welling. Semi-supervised learning with deep generative models. In Advances in Neural Information Processing Systems 27, pages 3581–3589. Curran Associates, Inc., 2014. [2] Eric Nalisnick and Padhraic Smyth. Stick-breaking variational autoencoders. International Conference on Learning Representations (ICLR), Apr 2017. [3] Francisco R Ruiz, Michalis Titsias RC AUEB, and David Blei. The generalized reparameteriza- tion gradient. In D. D. Lee, M. Sugiyama, U. V. Luxburg, I. Guyon, and R. Garnett, editors, Advances in Neural Information Processing Systems 29, pages 460–468. Curran Associates, Inc., 2016. [4] Figurnov, Michael, Mohamed, Shakir, and Mnih, Andriy. Implicit reparameterization gradients. Neurips, 2018

Reviewer 3



This is a very well written paper and enjoyed reading it. The motivation is clear, the authors describe problems that arose with the Dirichlet prior in a VAE setting and how they have been circumvented in reference 12. They propose that circumvention was necessary because of the otherwise intractable inference procedure. Being able to utilize the reparameterization trick und thus cheaply arrive at gradient updates enables them to recreate the described model in a more principled and rigorous way. They describe a stick-breaking construction for the new type of simplex distribution and how this initially leads to strong dependence on component ordering. Their solution to this is both straight forward and compelling. They conclude with thorough experiments, demonstrating the usefulness of their approach and showing the benefit of having access to closed form gradient updates in terms of smaller error rates. In all, I have very little to criticize, this is well executed and presented research and may have a large impact on VAE applications that are henceforth not limited to Dirichlet priors when modelling multivariate random variables.

[Author Response · NeurIPS 2019]

We thank all reviewers for their excellent feedback. Reviewer 2 noted the absence of an important baseline–using
the Kumaraswamy stick-breaking construction with fixed and constant ordering [3]–making it impossible to assess
the impact of our work. Our approximate integration over orderings was paramount to semi-supervised learning
performance (so much so we overlooked its inclusion). We augmented our results to include some of the requested
baselines. Unfortunately, recovering the random number seeds used in our initial experiments was not possible.
Therefore, the results for these new baselines (marked with [†]) will have experienced different initializations and data
folds than our original results. However, the test set, from which results are collected, is the same for ALL experiments.
Reviewer 2 noted that our tables seemed incomplete. Log likelihood (for unlabeled test data) was not available for our
M2 implementation nor the cited models. For clarity, we bifurcated the tables.

Table 1: Classification errors.

| Experiment | Method | Error | $p$-value |
|---|---|---|---|
| MNIST | MV-Kum. | $0.103 \pm 0.011$ | $--$ |
| 10 trials | IRG[†][1] | $0.100 \pm 0.009$ | $0.51$ |
| 600 labels | Kumar-SB[†][3] | $0.247 \pm 0.012$ | $1.97 \times 10^{-16}$ |
| $\dim(z) = 0$ | Softmax | $0.100 \pm 0.010$ | $0.48$ |
| MNIST | MV-Kum. | $0.049 \pm 0.005$ | $--$ |
| 10 trials | IRG[†][1] | $0.042 \pm 0.004$ | $2.67 \times 10^{-03}$ |
| 600 labels | M2 (ours) | $0.102 \pm 0.011$ | $4.17 \times 10^{-11}$ |
| $\dim(z) = 2$ | Kumar-SB[†][3] | $0.143 \pm 0.015$ | $4.44 \times 10^{-13}$ |
| | Softmax | $0.053 \pm 0.004$ | $0.08$ |
| MNIST | MV-Kum. | $0.018 \pm 0.002$ | $--$ |
| 10 trials | IRG[†][1] | $0.020 \pm 0.004$ | $0.30$ |
| 600 labels | M2 (ours) | $0.020 \pm 0.001$ | $0.03$ |
| $\dim(z) = 50$ | Kumar-SB[†][3] | $0.077 \pm 0.012$ | $8.73 \times 10^{-12}$ |
| | Softmax | $0.020 \pm 0.002$ | $0.01$ |
| SVHN | MV-Kum. | $0.288 \pm 0.025$ | $--$ |
| 4 trials | IRG[†][1] | $0.291 \pm 0.017$ | $0.85$ |
| 1000 labels | M2 (ours) | $0.396 \pm 0.010$ | $1.86 \times 10^{-04}$ |
| $\dim(z) = 50$ | Kumar-SB[†][3] | $0.707 \pm 0.012$ | $8.10 \times 10^{-08}$ |
| | Softmax | $0.332 \pm 0.009$ | $0.02$ |

Table 2: Log likelihoods.

| Experiment | Method | Log Likelihood | $p$-value |
|---|---|---|---|
| MNIST | MV-Kum. | $-6.2 \pm 5.9$ | $--$ |
| 10 trials | IRG[†][1] | $-6.1 \pm 6.1$ | $0.97$ |
| 600 labels | Kumar-SB[†][3] | $-6.4 \pm 6.2$ | $0.94$ |
| $\dim(z) = 0$ | Softmax | $-7.2 \pm 6.0$ | $0.71$ |
| MNIST | MV-Kum. | $45.19 \pm 0.47$ | $--$ |
| 10 trials | IRG[†][1] | $45.42 \pm 0.45$ | $0.28$ |
| 600 labels | Kumar-SB[†][3] | $45.30 \pm 0.41$ | $0.56$ |
| $\dim(z) = 2$ | Softmax | $44.19 \pm 1.25$ | $0.03$ |
| MNIST | MV-Kum. | $116.21 \pm 0.38$ | $--$ |
| 10 trials | IRG[†][1] | $116.76 \pm 0.39$ | $4.51 \times 10^{-03}$ |
| 600 labels | Kumar-SB[†][3] | $116.42 \pm 0.40$ | $0.22$ |
| $\dim(z) = 50$ | Softmax | $115.17 \pm 0.44$ | $2.36 \times 10^{-05}$ |
| SVHN | MV-Kum. | $669.69 \pm 0.37$ | $--$ |
| 4 trials | IRG[†][1] | $668.93 \pm 0.53$ | $0.06$ |
| 1000 labels | Kumar-SB[†][3] | $669.03 \pm 0.43$ | $0.06$ |
| $\dim(z) = 50$ | Softmax | $669.55 \pm 0.11$ | $0.49$ |

Figure 1: Variance of the ELBO's gradient's first dimension for GRG [4], RSVI [2], IRG [1], and MVK (ours) when fitting a variational posterior to Categorical data with 100 dimensions and a Dirichlet prior. They fit a Dirichlet. We fit a MV-Kumaraswamy.

The drop in classification performance for Kumar-SB [3] is due to the over allocation of probability mass to the final stick during sampling (Fig. 1 of our initial submission)–when the class-assignment posterior has high entropy, the fixed order Monte Carlo integration will bias the last label dimension. We too have included the comparison to Implicit Reparameterization Gradients (IRG) [1]. Here, we set $q(\pi; \alpha_\phi(x)) = \text{Dirichlet}(\pi; \alpha_\phi(x))$ in our semi-supervised model with the same architecture and compute gradients according to [1]. IRG's classification performance is similar to ours and not statistically distinguishable–except for the MNIST $\dim(z) = 2$ experiment, where it likely experienced luckier randomness. Deep learning frameworks' (e.g. TensorFlow, PyTorch, Theano, CNTK, MXNET, Chainer) distinct advantage is NOT requiring user-computed gradients. IRG uses independent Gamma samples to construct Beta and Dirichlet samples. IRG's principle contribution for gradient reparameterization is that it side-steps the need to invert the standardization function (i.e. the CDF). However, IRG still requires Gamma CDF gradients w.r.t. the variational parameters. These gradients do not have a known analytic form and therefore [1] applies forward-mode automatic differentiation to a numerical method. We argue this implementation is not straightforward for the common practitioner. Furthermore, implementing IRG, requires additional (often non-trivial) code to supply IRG gradients to the framework's optimizer. Conversely, our method has analytic gradients, enabling easy integration into ANY deep learning framework. To the best of our knowledge, IRG for the Gamma, Beta, and Dirichlet distributions only exists in TensorFlow (IRG was developed at Deep Mind). It is well established that gradient computations with lower variance are better [1, 2, 4]. We compare our method's gradient variance to these other methods in fig. 1. Reviewer 2, we did not test Gamma-SB [3] as they demonstrate inferiority to Kumar-SB (in the context of their non-parametric model's reconstruction error). Reviewer 1 is correct regarding $\pi$ and $x$ in lines 173 & 181. Reviewer 1, We agree our KL-Divergence discussion is not clear. Let $\alpha_\phi^{(j)}(x)$ be the $j^{th}$ concentration parameter of the inference network, and $\alpha^{(j)}$ be $j^{th}$ parameter of the Dirichlet prior. Then, for the set of all orderings $O$, $D_{KL}(q(\pi; \alpha_\phi(x)) \,||\, p(\pi; \alpha)) =$

$$D_{KL}(\text{MV-Kumaraswamy}(\alpha_\phi(x)) \,||\, \text{Dirichlet}(\alpha)) = \mathop{\mathbb{E}}_{o \sim \text{Uniform}(O)} \left[ \sum_{i=1}^{K-1} D_{KL}\left( \text{Kumaraswamy}\left(\alpha_\phi^{(o_i)}(x), \sum_{j=i+1}^{K} \alpha_\phi^{(o_j)}(x)\right) \,||\, \text{Beta}\left(\alpha^{(o_i)}, \sum_{j=i+1}^{K} \alpha^{(o_j)}\right) \right) \right]$$

We compute $D_{KL}(\text{Kumaraswamy}(a, b) \,||\, \text{Beta}(a', b'))$ analytically as in [3] with a $5^{th}$ order Taylor approx. We
suspect approximate symmetry arises with $O(K^2)$ samples vs. $K!$ for full symmetry. Since the problematic bias occurs
on the last stick, each label needs an ordering where it is not last, which has probability $K = \frac{K!}{(K-1)!}$ of occurring. Thus,
for this to occur for all labels, we have $K^2$. Reviewer 2, we set $K = 10$, the number of classes–not 50 as you suggest.
Reviewer 2 notes we could replace the Dirichlet prior with a MV-Kumaraswamy prior–we agree, however, chose the
Dirichlet prior for reader familiarity and because we are proposing a Dirichlet surrogate. Reviewer 2, if allowed, we
would be happy to change the title to "A New, Explicitly Reparameterizable Surrogate for the Dirichlet."

# References

[1] Mikhail Figurnov, Shakir Mohamed, and Andriy Mnih. Implicit reparameterization gradients. In *Advances in Neural Information Processing Systems 31*, pages 441–452. 2018.
[2] Christian A. Naesseth, Francisco J. R. Ruiz, Scott W. Linderman, and David M. Blei. Reparameterization gradients through acceptance-rejection sampling algorithms. In *Proceedings of the*
*20th International Conference on Artificial Intelligence and Statistics*, 2017.
[3] Eric Nalisnick and Padhraic Smyth. Stick-breaking variational autoencoders. *International Conference on Learning Representations (ICLR)*, Apr 2017.
[4] Francisco R Ruiz, Michalis Titsias RC AUEB, and David Blei. The generalized reparameterization gradient. In D. D. Lee, M. Sugiyama, U. V. Luxburg, I. Guyon, and R. Garnett, editors,
*Advances in Neural Information Processing Systems 29*, pages 460–468. Curran Associates, Inc., 2016.


[Meta-Review · NeurIPS 2019]

After the rebuttal there is consensus among reviewers that the paper should be accepted. The authors provided very convincing feedback in their rebuttal response, including experiments with new baselines, and are strongly encouraged to supplement the paper with these (as we're sure they would)! The reviews for this paper were very detailed, and there is not much that a meta-summary could add. On a technical note, the authors could supplement their explanation somewhat; in that readers could "see that the density p(o) of the orderings can be taken out of the KL terms, but it is not clear why the dependence of x_{o_i} on x_{o_1}, ...,x_{o_{i-1}} seems to not matter in the KL-calculation", to quote verbatim.